# Multigenerational Effects of Graphene Oxide Nanoparticles on *Acheta domesticus* DNA Stability

**DOI:** 10.3390/ijms241612826

**Published:** 2023-08-15

**Authors:** Barbara Flasz, Amrendra K. Ajay, Monika Tarnawska, Agnieszka Babczyńska, Łukasz Majchrzycki, Andrzej Kędziorski, Łukasz Napora-Rutkowski, Ewa Świerczek, Maria Augustyniak

**Affiliations:** 1Institute of Biology, Biotechnology and Environmental Protection, University of Silesia in Katowice, 40-007 Katowice, Poland; barbara.flasz@us.edu.pl (B.F.);; 2Department of Medicine, Division of Renal Medicine, Brigham and Women’s Hospital, Harvard Medical School, Boston, MA 02115, USA; 3Center for Advanced Technology, Adam Mickiewicz University, 61-614 Poznań, Poland; 4Polish Academy of Sciences, Institute of Ichthyobiology and Aquaculture in Gołysz, 43-520 Chybie, Poland

**Keywords:** graphene oxide, multigenerational effects, oxidative stress, DNA damage, DNA methylation, epigenetics, invertebrate, nanotoxicity, 8-OHdG, AP sites

## Abstract

The use of nanoparticles like graphene oxide (GO) in nanocomposite industries is growing very fast. There is a strong concern that GO can enter the environment and become nanopollutatnt. Environmental pollutants’ exposure usually relates to low concentrations but may last for a long time and impact following generations. Attention should be paid to the effects of nanoparticles, especially on the DNA stability passed on to the offspring. We investigated the multigenerational effects on two strains (wild and long-lived) of house cricket intoxicated with low GO concentrations over five generations, followed by one recovery generation. Our investigation focused on oxidative stress parameters, specifically AP sites (apurinic/apyrimidinic sites) and 8-OHdG (8-hydroxy-2′-deoxyguanosine), and examined the global DNA methylation pattern. Five intoxicated generations were able to overcome the oxidative stress, showing that relatively low doses of GO have a moderate effect on the house cricket (8-OHdG and AP sites). The last recovery generation that experienced a transition from contaminated to uncontaminated food presented greater DNA damage. The pattern of DNA methylation was comparable in every generation, suggesting that other epigenetic mechanisms might be involved.

## 1. Introduction

Graphene oxide (GO) is an oxidized graphene derivative with a two-dimensional structure [1]. Its structure consists of hexagonal rings accompanied by various oxygen-rich functional groups, including hydroxyl (-OH), epoxy (C-O-C), carbonyl (C=C), or carboxyl (-COOH), among others. These functional groups determine the particle’s polarity [2,3] and influence GO’s solubility in water. This solubility advantage enables surface functionalization, making it a desirable feature in the production of nanocomposites for various industries [2]. GO finds applications in several industries, such as electronics, textiles, and medicine, particularly in diagnostics [4,5,6,7,8]. However, the widespread use of graphene oxide in various industries raises concerns about the potential release of these nanoparticles into the environment. Therefore, understanding the biological consequences of exposure to nanoparticles in the environment, even at trace concentrations, is of utmost importance.

Previous studies have demonstrated that graphene oxide intoxication can lead to various disorders, affecting development, reproduction, protein activity, and DNA stability within a single generation that was investigated [9,10,11,12,13,14]. However, knowledge of GO under continuous exposure remains limited. The presence of diverse nanoparticles (NPs) in food and the environment is widely acknowledged [15,16,17,18,19]. Thus, a significant question arises regarding the impact of NPs on organisms and their offspring, which experience continuous exposure to NPs, even at low doses. The importance of conducting multi- or trans-generational studies on the effects of stressors has recently been increasingly emphasized [20,21]. To date, the vast majority of published findings have focused on the multigenerational imprint of NPs on developmental and reproductive parameters [21,22,23,24,25,26]. However, limited attention has been given to studying the multigenerational effects of GO treatment, including oxidative DNA damage and epigenetic changes [15,27,28,29,30,31,32,33]. In a study by Wang and Liu [15], the transgenerational effects of silver NPs on the gastropod *L. stagnalis* were investigated. Parental exposure to silver NPs increased ROS production in both parents and offspring (eggs). Additionally, multigenerational exposure to TiO_2_ NPs led to elevated intracellular reactive oxygen species (ROS) production and upregulation of antioxidative genes [29]. In the other studies, *C. elegans* was subjected to multigenerational exposure to silver NPs. These investigations focused on examining epigenetic changes, particularly alterations in histone methylation markers and the expression of genes encoding demethylases and methyltransferases associated with the selected markers. In both parameters, the multigenerational exposure to silver NPs induced epigenetic changes that were inherited by subsequent unexposed offspring [30]. Furthermore, exposure to copper oxide NPs for five generations, followed by two generations with clean media, resulted in increased global DNA methylation, which corresponded to phenotypic effects such as changes in reproduction. Additionally, gene expression analysis revealed epigenetic changes in stress and detoxification genes that varied depending on the generation and form of NPs used. These observed effects were detected in post-exposure generations, indicating transgenerational effects [33].

Given the promising properties of GO that make it suitable for various industrial applications, we investigated the multigenerational effects of GO exposure on DNA stability parameters. To our knowledge, limited studies have examined DNA stability under multi- or trans-generational exposure to graphene oxide. DNA is a crucial molecule, and its damage can profoundly affect a species. Therefore, it is important to assess the impact of multigenerational GO exposure on subsequent generations, particularly regarding DNA damage and epigenetic patterns. In this project, we investigated the multigenerational effects on two strains of the invertebrate *Acheta domesticus* intoxicated with low GO concentrations over five generations, followed by one recovery generation. These strains were chosen due to their distinct ontogenetic development, making them attractive for research. Our investigation focused on oxidative stress parameters, specifically AP sites (apurinic/apyrimidinic sites) and 8-OHdG (8-hydroxy-2′-deoxyguanosine). Additionally, considering the evidence suggesting that engineered nanomaterials can induce epigenetic modifications that can be passed to the offspring [34], we examined the pattern of global DNA methylation in both strains. We believe that the presented results can contribute to the existing knowledge in this field.

## 2. Results

### 2.1. Graphene Oxide 

The SEM and AFM images revealed mainly single-layered GO flakes with various dimensions (up to several micrometers) and a flake thickness of approximately 1.0 nm (Figure 1a,b). The average flake area was about 2 µm^2^. The zeta potential for the sample was measured to be −28.8 mV, confirming the excellent stability of the suspension (Figure 2). A more detailed description of the sample, including XPS analysis, can be found in our previous work, where the Nanografi sample was labeled S3 [35].

This project examined two parameters of DNA damage, namely AP sites (apurinic/apyrimidinic sites) and 8-OHdG (8-hydroxy-2′-deoxyguanosine), as well as the level of global DNA methylation. Statistical analysis using ANOVA/MANOVA was conducted for each investigated main factor, which included the effects of generation (G), treatment (T; control and GO groups), and strain (S; wild and long-lived) separately, as well as the interactions between the factors (their combined impact on the dependent variables). Multivariate analysis of variance (MANOVA), considering all three dependent variables simultaneously, revealed that factors G and strain S significantly influenced these parameters, while factor T did not have a significant impact. However, further analysis revealed the presence of meaningful interactions between the factors G × T, T × S, and G × T × S (Table 1). This means that although the overall treatment with low doses of GO did not significantly affect the examined parameters in *A. domesticus* (considering all animals investigated during the whole experiment), differences were possible in the GO-treated groups across generations and strains. Therefore, in the next step, the results were analyzed separately for each parameter, thoroughly examining the impact of factors G, T, and S.

### 2.2. DNA Damage: AP Sites (Apurinic/Apyrimidinic Sites)

In six generations of two A. domesticus strains, apurinic/apyrimidinic DNA lesions were measured. No statistically significant differences were observed in generations G1 to G5 for both strains, considering the control and GO-treated groups. Interestingly, in the last recovery generation of insects, significant differences were observed between treated groups in the wild (H) and long-lived (D) strains (Figure 3a and Figure 4a). The results showed higher levels of AP sites in groups fed food with a higher GO concentration compared to the control group.

Despite the lack of significant differences in G1 to G5 in the wild-type strain, the trends were visible (Table 2a). The first generation (G1), exposed to the higher GO concentration in their food, presented slightly higher levels of abasic sites in the DNA than the control group. In G2, the levels of AP sites in the control group and the lower and higher GO-treated groups were very similar. Generations 3–5 exhibited lower AP site lesions in both variants of GO-treated groups compared to the control group. In the last G6 generation (recovery), all groups were fed control food without GO. In that case, higher levels of AP sites were observed in the lower and higher GO-treated groups compared to the control group. The increasing level of DNA damage was correlated with the past treatment (lower or higher GO concentration in food) of crickets. Cricket animals fed with higher GO concentrations in previous generations exhibited more pronounced AP site lesions.

In the long-lived strain, trends were also noticed (Table 3a). Fluctuations were noticeable, with alternating increases and decreases in AP site lesions in successive generations. In G1 and G2, the lower GO-treated group exhibited a slightly higher AP site level than the control. The higher GO-treated group showed levels almost similar to the control. The third generation of intoxicated insects displayed slightly lower AP site lesions compared to the control. In the G4, the AP sites were higher in the GO-treated groups compared to the control group. Notably, G4, including the control group, had the lowest number of detected AP sites among all generations. In the last GO-treated generation, G5, the levels of abasic sites were smaller than in the control group. In the recovery generation (G6) of the D strain, the same trends in AP site lesions were observed as in the wild-type insects. Higher AP site lesions were observed in the lower and higher GO-treated groups compared to the control group, and the increase in lesions was correlated with the treatment.

### 2.3. DNA Damage: 8-OHdG (8-hydroxy-2′-deoxyguanosine)

8-hydroxy-2′-deoxyguanosine (8-OHdG) is a predominant form of oxidative damage induced by free radicals (Figure 5). This study employed this oxidative stress biomarker to assess oxidative stress in the gut of wild and long-lived strains of house crickets over six consecutive generations. The results are presented in Figure 3b and Figure 4b.

Significant differences were observed in the wild strain during G1 and G2 (Figure 3b). In the first generation, exposed to two different GO concentrations in their food, the group treated with the lower GO concentration exhibited a high level of significant 8-OHdG damage. Conversely, the higher GO-treated group presented lower DNA oxidative damage than the control and lower GO-treated groups. In the second generation, the lower GO-treated group displayed the lowest DNA lesions, which was statistically significant compared to the control and higher GO-treated groups. From G3 to G6, no statistically significant differences were found, but trends were observed (Table 2b). In generations 3 and 4, oxidative stress levels were similar in the control, lower, and higher GO-treated groups. G5 presented lower DNA damage in both the lower and higher GO-treated groups compared to the control. In the last (recovery) generation 6, no significant differences were observed between the groups, with similar results for 8-OHdG.

The long-lived strain seemed to be more sensitive to the effects of GO in food, which was manifested by more significant differences in 8-OHdG in the D strain compared to the H strain (Figure 4b). In the first generation of investigated insects, both the lower and higher GO-treated groups exhibited higher levels of 8-OHdG DNA lesions than the control. Moreover, a higher concentration of GO in the food corresponded to a higher level of 8-OHdG, with statistical significance in both treated groups. In G2, the higher GO-treated group showed significantly lower DNA damage compared to the control. In the G3, both lower and higher GO-treated groups presented significant differences compared to the control group, showing fewer DNA lesions. G4 and G5 did not reveal any significant differences among all groups. In the last generation, which was not fed GO in their food, the results showed significant differences in the higher GO-treated group compared to the control group. It is worth mentioning that G1, G4, and G6 presented a similar, concentration-dependent pattern in the level of the 8-OHdG biomarker. The higher the GO concentration in the food, the more significant DNA damage occurred (Figure 4b and Table 3b). Overall, the trends indicated more elevated DNA lesions caused by GO treatment in G1, G4, and G6. GO-treated insects from generations G2, G3, and G5 showed lower or similar levels of DNA damage compared to the control group. Thus, a fluctuation among generations was observed regarding 8-OHdG, including in the control groups.

### 2.4. Global DNA Methylation

The last parameter investigated was global DNA methylation, which represented epigenetic changes in six generations of crickets. Overall, the level of global DNA methylation was relatively low, with an average of no more than 5% in the investigated groups.

In the wild strain, significant differences were observed only in generation 5, where the group treated with the lower GO concentration in their food showed lower levels of methylated DNA compared to the control (Figure 3c). Although only a few significant differences were detected, trends were noted in the H strain (Table 2c). In the first generation, global methylation was similar in the control and insects treated with a higher concentration of GO. However, the lower GO-treated group exhibited lower DNA methylation levels. In G2, both the control and higher GO-treated groups had similar DNA methylation levels, while the lower GO-treated group showed slightly higher global DNA methylation. In G3, the higher the GO concentration was, the higher DNA methylation occurred. In generation 4, both lower and higher GO-treated groups had lower DNA methylation compared to the control group. G5 exhibited lower DNA methylation in both the lower and higher GO-treated groups compared to the control group. The last recovery generation (6) generally displayed the lowest level of global DNA methylation. The higher GO-treated group had a similar level of global DNA methylation to the control group. In comparison, the lower GO-treated group presented a higher methylated DNA level than the other two groups. Once again, intergenerational fluctuations in this parameter were observed.

In the long-lived strain, no significant differences were found in the results of the six subsequent generations (Figure 4c). Despite the lack of statistical differences, interesting trends were observed. The highest global DNA methylation was observed in generation 1 for all investigated groups. Generations 2 and 3 presented relatively low DNA methylation levels. The level of methylated DNA increased again in generations 4 and 5. In generation 6, when GO-spiked food was no longer provided, all the groups exhibited the lowest DNA methylation levels among all generations investigated.

The general trends in the D strain in generation 1 showed lower DNA methylation in GO-treated groups than in the control group (Table 3c). The lower GO-treated groups in G2 and G3 showed slightly lower DNA methylation than the control group, while the higher GO-treated group presented higher DNA methylation. In G4, both investigated groups presented higher DNA methylation than the control. However, in G5, the pattern reversed, with lower DNA methylation observed in both the lower and higher GO-treated groups compared to the control. In generation 6 (recovery), the lower GO-treated group presented a higher global DNA methylation level than the control, and the higher GO-treated group exhibited even higher global DNA methylation than the control and the lower GO-treated groups.

It is worth mentioning that in a long-lived strain, generation six displayed a similar pattern in all investigated parameters: AP sites, 8-OHdG DNA, and global DNA methylation. The increase in the value of the measured parameter corresponded to a rise in the concentration of GO in the food that ancestors were fed (Figure 4a–c).

### 2.5. PCA—Relationships between Variables

Principal Component Analysis (PCA) was performed separately for each strain and experimental group to reveal possible relationships between investigated parameters (Figure 6). Both principal components (PC1 + PC2) explained a significant percentage of the variability in both the H and D insect strains. In the H strain (wild), both components explained 76.59% in the control group, 69.38% in the lower group, and 84.21% in the higher group. In the long-lived (D) strain, in the control, lower, and higher groups, the two principal components explained 83.94%, 81.99%, and 76.23% of the overall data variability, respectively.

In the H strain, PC1 was created by AP site parameters and 8-OHdG, revealing negative correlations between these two parameters in the control and lower groups and a positive correlation in the higher group. Global DNA methylation (represented by PC2) showed no correlation with AP sites or 8-OHdG. Generally, PCA analysis in the H strain showed a similar pattern for the control and lower groups, indicating that the low concentration of GO did not affect the measured parameters. However, exposure to the higher concentration of GO caused a rearrangement in the 8-OHdG parameter.

In the D strain, PC1 was created mainly by AP site parameters, while the other parameters were partly related to PC1 and PC2. In the case of the D strain, both lower and higher concentrations influenced the relationships among the measured parameters, resulting in a change in the arrangement of parameters in a coordinate system. However, no strong correlations among parameters were evident (Figure 6).

## 3. Discussion

Although not as widely conducted as single-generation studies, multigenerational studies hold significant importance and relevance. The examination of successive populations over multiple generations can provide valuable insight into the nature of the parameter being investigated as well as the characteristics of the studied species or population. Comprehensive multigenerational studies have demonstrated adaptive or evolutionary changes within the studied species. These studies offer intriguing information regarding alterations in population size, reproductive rate, enzyme activity, cell size, and viability, as well as nuclear DNA changes that may be challenging to detect within a single generation [36,37,38,39,40]. Species constantly undergo changes, including those of a fluctuating nature, in an effort to adapt to their evolving needs. These ongoing modifications can serve as an initial stage of evolutionary processes [41,42]. Many wild populations experience multigenerational cycles due to factors such as predator-prey relationships or seasonal changes [43]. Despite maintaining constant conditions in the breeding room during experiments, fluctuations remain noticeable. For example, in a three-generation study involving *Acheta domestics* that were intoxicated with GO, the reproduction rate was found to be influenced by seasonal variations [11]. Another study involving multigenerational exposure of *Spodoptera exigua* to cadmium revealed fluctuations in DNA stability. The authors hypothesized that these fluctuations would persist as long as the population reached a satisfactory level of adjustment, ensuring survival or becoming extinct [44]. Our study provides further support for this thesis. Throughout six generations in the controlled breeding room with constant conditions, we observed noticeable fluctuations in the parameters under investigation. Specifically, regarding oxidative DNA damage (Figure 3 and Figure 4), the fluctuations were more pronounced in the long-lived strains. Conversely, fluctuations in global DNA methylation over generations were more significant in the wild-type strain. These differences could be attributed to the slightly different life strategies of the insect strains. The wild strain is characterized by a strategy focused on quickly producing many offspring. In contrast, the long-lived strain is oriented towards producing offspring over a more extended period.

Currently, most investigations involving multiple generations exposed to various NPs (such as gold, silver, cobalt, polystyrene, silicon dioxide, and quantum dots) have primarily focused on reproductive disorders [23,24,25,45,46,47,48]. For instance, in a study with *D. melanogaster*, the insects were exposed to different concentrations (1, 10, 50, and 100 mg∙L^−1^) of polystyrene nanoplastics over five generations. The findings revealed the accumulation of polystyrene in the crop, gut, and ovaries, leading to a decrease in egg production and hatching rate. In the fifth generation, oocytes revealed a significant level of apoptosis and necrosis, which correlated with dose. The study proposed a potential mechanism for reproductive toxicity [45]. On the other hand, Heinlaan et al. [46] demonstrated that consecutive 21-day exposure to polystyrene NPs at concentrations of 0.1 and 1 mg∙L^−1^ had no impact on the three investigated generations. Only in the third generation did the fertility rate increase. The cited works demonstrated that, in many cases, the adverse effects of risk factors are concentration- or dose-dependent. When low doses are employed, health or reproduction parameters may be comparable to those of unexposed controls. However, the question of bioaccumulation in subsequent generations arises, specifically whether the succeeding generations can effectively eliminate the toxic agent from their organisms. In other studies, the transgenerational effects of gold nanoparticles on offspring have been observed. Parental exposure to gold nanoparticles exhibited biological effects on the subsequent generation, leading to developmental toxicity [15]. Another investigation examining the effects of gold NPs and parental exposure revealed that the reproduction rate was affected in the second generation but recovered in the third and fourth generations [23].

Only a few multigenerational nanoparticle exposure studies have specifically investigated oxidative stress [15,27,28,29]. The production of reactive oxygen species (ROS) significantly impacts various aspects of an organism’s functioning, including survival, growth, and locomotion, but also inflammation, membrane damage, apoptosis, and genotoxicity [29]. The major oxidative lesions that affect DNA stability are AP sites and 8-OHdG [49,50,51,52]. Given the limited knowledge regarding DNA oxidative damage in multigenerational studies of NPs, we decided to examine AP sites (abasic) and 8-OHdG lesions as biomarkers of oxidative stress caused by GO.

Damaged DNA bases can result from various processes such as methylation, oxidation, and deamination [50]. Specific DNA *N*-glycosylases are responsible for removing most of the inappropriate bases, resulting in the formation of abasic sites [50,53,54]. While a certain amount of AP sites is tolerated in an organism, excessive accumulation of these lesions can lead to cytotoxicity by blocking DNA replication and transcription or even the formation of single or double-strand breaks (SSBs or DSBs) [49,50,53]. Maintaining DNA stability relies on a delicate balance between the formation of AP sites and the activity of molecular repair mechanisms such as base excision repair (BER), nucleotide excision repair (NER), and recombination. Our study involving five generations exposed to GO did not reveal a significant formation of AP sites. Both strains, in both controls and GO groups, exhibited a fluctuating pattern in the AP site parameter (Figure 3a and Figure 4a). The presence of GO in low concentrations did not alter this pattern, suggesting that the balance between AP site formation and DNA repair mechanisms was likely maintained or that the concentrations used in the experiment were relatively safe and harmless. However, an interesting exception was observed in the recovery generation (G6). It appears that removing GO from food after five generations of exposure posed a greater burden than constant and continuous intoxication. This withdrawal, or perhaps a change in conditions, seems to have led to an increase in the formation of more abasic sites. Both the wild strain and long-lived strain exhibited a similar response, indicating a relationship to GO concentration in previous generations. However, the established balance between AP site formation and DNA repair during GO exposure was disturbed under the new conditions, requiring the organism to allocate energy costs to develop a new equilibrium. The question arises as to how long it takes to restore homeostasis. In the study conducted by Kim et al. [23], it took three generations of *D. melanogaster* to recover after parental exposure to gold NPs. Another study investigated the multigenerational effects of parental exposure to different carbon NPs (fullerenes, single-walled carbon nanotubes, and multi-walled carbon nanotubes). It concluded that the surface chemistry of the nanomaterial influences its toxicity, resulting in decreased survival or reproduction in the offspring (F1 and F2) [55]. Undoubtedly, the effect of toxicity or the ability to return to homeostasis depends on various factors such as the type of nanoparticles, concentration or dose, route of administration, and species used in the experiment.

8-OHdG (8-hydroxy-2′-deoxyguanosine) is a well-established biomarker of oxidative stress [51,52,56,57,58]. It is a metabolite of oxidative stress produced in the reaction of hydroxyl radicals with DNA strands [51,52]. 8-OHdG indicates an imbalance between ROS production and the efficiency of scavenging systems. Given that the main proposed mechanism of GO action is the generation of oxidative stress [4,59,60,61,62], measuring 8-OHdG serves as a suitable choice due to its high sensitivity to oxidative DNA damage. Most of the 8-OHdG biomarker studies are single-generation studies conducted in vertebrates, with a limited number of successive studies in invertebrates [63,64,65,66,67,68]. Multigenerational or transgenerational studies considering the 8-OHdG are relatively scarce. For instance, in sea urchins, the effect of maternal exposure to polycyclic aromatic hydrocarbons was examined. The activity of antioxidative enzymes, oxidative lipid and protein damage, and biomarkers of oxidative stress, including 8-OHdG, were measured in gonads, eggs, and early-stage progeny. The study revealed increased levels of 8-OHdG in eggs and progeny, indicating elevated DNA damage [69]. 8-OHdG in the wild strain and the long-lived strain showed greater DNA damage than AP sites (Figure 3b and Figure 4b). The D strain appeared more susceptible to oxidative DNA damage, with significant differences observed in generations G1 to G3 and the recovery generation G6 (Figure 4b). On the other hand, the H strain displayed significant differences only in generations G1 and G2 (Figure 3b). These variations between the strains may be attributed to their different developmental strategies. When insects of the D strain were intoxicated with GO in generations G1, G2, and G3, noticeable differences in 8-OHdG levels were visible compared to the controls. In G1, continuous GO treatment led to a concentration-dependent increase in 8-OHdG in the long-lived strain. In G1 (H strain), G2, and G3 (D strain), although no clear relationship was established, the levels of 8-OHdG differed from the controls. Exposure to the risk factor likely disturbed the redox balance in generations G1 to G3, resulting in fluctuations in the 8-OHdG parameter. The fact that both strains in generations G4 and G5 presented levels of 8-OHdG similar to the control groups suggests a return to balance. It should be noted that the concentrations of GO used in the study were relatively low, indicating that they may not have caused harm to the investigated organism. Interestingly, in the D strain, significant differences were observed in the last recovery generation (G6). It is worth noting that in the recovery generation (G6) of the D strain, a similar pattern of DNA damage and AP sites was observed. This indicates a correlation between oxidative damage and GO concentration (which was consumed by their ancestors). These results provide further support for our supposition that in the last recovery generations, homeostasis was disturbed, and the elimination of GO posed greater stress than the consumption of food contaminated with GO.

DNA methylation serves as an epigenetic mark of cellular memory [70]. It is hypothesized that exposure to NPs can alter patterns of DNA methylation, posttranscriptional histone modifications, and expression of non-coding RNA, depending on the physiochemical proprieties as well as the cell’s or organism’s sensitivity [34]. Multi- or trans-generational exposure to NPs can lead to developmental changes in offspring. In insects, the percentage of methylated cytosines is generally low (0–3%), and their functional role is still poorly understood [71]. However, it is reported in many insect species [71,72]. In our study, we observed a generally low level of global DNA methylation. With the exception of the wild-type strain in generation G5, there were no significant differences in the level of methylation across subsequent generations (Figure 3c and Figure 4c). Similar to other parameters, we noted fluctuations in the methylation level over generations. Interestingly, in the last recovery generation (G6), both strains exhibited the lowest level of DNA methylation. In the D strain, we observed a relationship between GO concentration consumed by ancestors and methylation level in progeny, similar to other parameters such as AP sites and 8-OHdG (Figure 4). The lack of significant differences in DNA methylation across generations in our study is an intriguing finding that warrants further investigation. It suggests that methylation may not be the primary epigenetic mechanism in *A. domesticus*, and it highlights the need to explore other epigenetic mechanisms. For instance, in the case of *Enchytraeus crypticus* exposed to copper oxide over five generations, no significant differences were detected in DNA methylation. This led the authors to propose RNAi as a potential epigenetic mechanism [33]. Similarly, no significant changes in global DNA methylation levels were observed in a study involving *C. elegans* exposed to silver NPs for three generations, followed by two generations of recovery. However, when the nematodes were intoxicated with AgNO_3_, an increase in global DNA methylation was detected. This led the authors to hypothesize the existence of different epigenetic marks for AgNPs and AgNO_3_ [31].

In conclusion, our study indicates that relatively low doses of GO have a moderate effect on the house cricket, and the insects appear capable of adapting to these ‘inconveniences.’ In many cases, the adverse effects of risk factors depend on their concentration or dose. When low doses are utilized, the health or reproduction parameters may resemble those of the controls that were not exposed. The stress factor presented by GO interacts with the inherent biological fluctuations of various parameters within the species, leading to slight modifications. Notably, not toxic but noticeable effects appear in the generations that experience a transition from contaminated to uncontaminated food conditions. On the other hand, one should remember that non-toxic agents may pose a risk in different conditions. While GO may not be toxic in low doses, it can serve as a drug carrier or combine with toxic or risky substances that will be transferred to subsequent generations.

## 4. Materials and Methods

### 4.1. Graphene Oxide Characteristics

Graphene oxide (GO) was supplied by Nanografi, Germany, as an aqueous suspension at a concentration of 10 mg∙mL^−1^. Before use, the material was visualized using SEM and AFM techniques, and the stability of the suspension was assessed by measuring the zeta potential. The GO dispersion was diluted to a concentration of 20 μg∙mL^−1^ and sonicated for 1 min using an ultrasonic bath. It was then deposited on a silicon wafer for SEM measurements and freshly cleaved mica for AFM measurements. The samples were left to dry overnight at room temperature. The morphology of the samples was examined using a scanning electron microscope (SEM) Quanta FEG 250 (FEI, Oregon, USA) operating at an accelerating voltage of 30 kV in high vacuum mode. AFM imaging was performed using an Agilent 5500 atomic force microscope (Agilent Technologies, CA, USA) in tapping mode. The zeta potential was measured at 25 °C using a Litesizer 500 (Anton Paar, Graz, Austria), and the zeta potential values were calculated using the Smoluchowski equation.

### 4.2. Characteristics of the Species

*Acheta domesticus* (*Gryllidae, Orthoptera, Insecta*) is a worldwide species. It has many advantages that make this medium-sized insect attractive for research as a model organism. The house cricket life cycle lasts about three to four months with a high reproduction rate, facilitating the design of the desired number of investigation groups with sufficient repeats. Breeding is relatively easy, and the insect is omnivorous [73]. The Institute of Biology, Biotechnology, and Environmental Protection at the University of Silesia in Katowice has been conducting unique selective breeding of *Acheta domesticus*. The breeding results are two strains that differ in ontogenetic development [9,10,26]. This project used the wild-type strain (H) and the long-lived strain (D). The D strain has a longer ontogenetic development time and a longer life expectancy than the H strain [10].

### 4.3. Food Preparation: Graphene Oxide, Control Food

Graphene oxide food was prepared from standard artificial food (Kanisan Q, Sano, Poland). The food pellet was ground with a laboratory mill. The next step was mixing GO dissolved in ultrapure water. Two GO concentrations were prepared: lower (0.02 mg∙kg^−1^ of dry food) and higher (0.2 mg∙kg^−1^ of dry food). Then, the GO food was dried (24 h, 45 °C) and sterilized with a UV lamp (24 h). Dry food was kept in sealed plastic bags and used during the experiment. This same protocol was used for preparing control food, but water was added instead of GO.

### 4.4. Experimental Model and Tissue Preparation

The insects were kept in a laboratory breeding room with optimal conditions for reproduction and development. The condition in the breeding room was as follows: temperature 28.8 ± 0.88 °C; photoperiod L:D 12:12; humidity 20–45%. Two populations of *A. domesticus* were used for the project: wild (H) and long-lived (D) (Figure 7). One-week-old larvae were divided into experimental groups (control, lower, and higher) and placed in the insectaria. In total, six groups were created. Crickets were fed *ad libitum* with water and GO-contaminated or controlled food for their whole life. Five insects were collected and sacrificed for future analysis on the fifteenth day of Imago’s life. Gut tissue was taken and placed in a 1.5 mL tube with RNAlater (Merck, Germany) and frozen at −80 °C until measurements. At the same time, some of the insects from each research group were intended for reproduction to create the next generation (two strains, three experimental groups each). The process was repeated until the last sixth generation. As the last G6 was named recovery, the treatment groups were created, but all groups were fed control food that was not GO-contaminated.

### 4.5. DNA Isolation

DNA was isolated with a commercially available Genomic Mini Kit (A&A Biotechnology, Gdansk, Poland). The samples were slowly defrosted on ice, and RNAlater was removed. The gut tissue was rinsed twice with Tris buffer (Genomic Mini Kit), homogenized, and the protocol for DNA isolation was followed. DNA was suspended in 100 µL of Tris buffer. The DNA quality and concentration were measured (NanoDrop 2000, Thermo Fisher Scientific, Waltham, MA, USA), and it was stored at −80 °C to analyze DNA global methylation, AP sites, and 8-OHdG.

### 4.6. Measurement of Selected Parameters

#### 4.6.1. AP Sites (Apurinic/Apyrimidinic Sites)

DNA was slowly defrosted on ice, and the DNA Damage Quantification Colorimetric Kit (Abcam, Cambrige, UK) was used to process the samples. The DNA was diluted to 0.1 µg∙µL^−1^ concentration in every sample. The manufacturer’s protocol was followed, and the DNA AP site lesions were calculated using a standard curve.

#### 4.6.2. 8-OHdG (8-hydroxy-2′-deoxyguanosine)

DNA was defrosted on ice. Samples were diluted to 200 ng each in 50 µL of assay buffer. Then, the enzyme-linked immunosorbent assay using the DNA Damage Competitive ELISA Kit (Invitrogen, Carlsbad, CA, USA) was performed. The 8-OHdG oxidative damage in DNA was calculated based on the standard curve.

#### 4.6.3. Global DNA Methylation

For measuring global DNA methylation, DNA was defrosted on ice and diluted to 100 ng in an assay buffer. The Global DNA Methylation Assay Kit (Abcam) was used to process the samples. Standard curves and instructions from the protocol were used to calculate global methylation levels.

### 4.7. Statistical Analysis

Statistical analysis was performed using professional software (STATISTICA^®^13.3 PL, StatSoft Polska, Kraków, Poland). For all the parameters: AP sites, 8-OHdG, and global DNA methylation, five repeats in every treatment group were analyzed (*n* = 5). Additionally, in 8-OHdG, four technical repetitions were made. On AP sites, three technical repetitions were made. The Kolmogorov–Smirnov and Shapiro–Wilk tests were used to determine the normality of the processed data. Due to the non-normal distribution of all measured parameters, the Box–Cox transformation was used to transform the data into a normal distribution. The homogeneity of the variances was verified by the Levene and Brown–Forsythe tests. The transformed data fulfilled the analysis of variance criteria. Two-way ANOVA (post-hoc Fisher test, *p* < 0.05) was applied to assess the effects of independent variables on dependent variables. Also, MANOVA (Wilks’ Lambda test; *p* < 0.05) was used to determine the effects of all factors: strain, generation, and treatment. PCA (Principal Component Analysis) was used to evaluate the relationship among investigated parameters within strain and treatment groups. The Box–Cox transformed results shown in the figures and tables are represented as mean ± SE.

The null hypothesis that was tested during statistical analyses was formulated below:

**H1.0.** 
*The DNA damage in the groups exposed to GO is similar to a control group and does not reveal significant differences among generations.*


Rationale: The reaction to GO intoxication is related to the dose and is not passed on to the other generation exposed to the risk factor. Therefore, the doses used in the experiment should be considered low, safe for the organism, and not cause adverse effects in the tested generations.

**H1.1.** 
*The DNA damage is various in consecutive generations intoxicated with GO. Even low doses of GO consumed in food continuously over five generations impact DNA stability and lead to lower/higher DNA oxidative lesions. The level of DNA damage is related to the species’ strategy and/or the availability of molecular tools to combat oxidative stress in the organism.*


**H2.0.** 
*Global DNA methylation is comparable in consecutive generations intoxicated with GO. The low doses of GO consumed in food do not affect epigenetics; specifically, DNA methylation patterns and/or global methylation patterns are not sensitive enough parameters to detect subtle methylation changes.*


**H2.1.** 
*Global DNA methylation has a different pattern in the next generations of GO-intoxicated Acheta domesticus. The epigenetic changes are related to the constant impact of risk factors. The GO treatment impacts epigenetic memory changes, creating a unique DNA methylation pattern that regulates tissue-specific gene transcription. Even low doses of GO can modify the global methylation pattern.*


**H3.0.** 
*The sixth generation (recovery) does not differ in DNA stability parameters from other generations intoxicated with GO. Low doses of GO did not affect the exposed generations (G1–G5), so the stress caused by GO removal will also be negligible.*


**H.3.1.** 
*The sixth generation (recovery) differs in DNA stability parameters from other generations intoxicated with GO. Long-term GO exposure has an impact on shifts in energy costs. Thus, removing the risk factor disturbs the balance, and a new homeostasis must be developed.*


**H4.0.** 
*Selection for longevity does not change the investigated DNA parameters, and the strains react similarly to GO intoxication in food. The lack of significant differences may indicate that both strains of A. domesticus have similar mechanisms to maintain DNA stability that are equally important for both strains.*


**H4.1.** 
*Selection for longevity changes the investigated DNA parameters, and the strains react differently to GO intoxication in food. The long-lived strain may respond differently to oxidative stress than the wild-type. The over-generational strain-dependent response may result from various ways of developing the intracellular machinery to fight oxidative stress, including antioxidant enzymes, DNA repair mechanisms, or epigenetic processes that regulate gene transcription.*


Rejection of null hypotheses will confirm the significant impact of graphene oxide on the investigated parameters across strains and generations and the beneficial/unbeneficial effect of removing GO from food in the last generation.

## Figures and Tables

**Figure 1 ijms-24-12826-f001:**
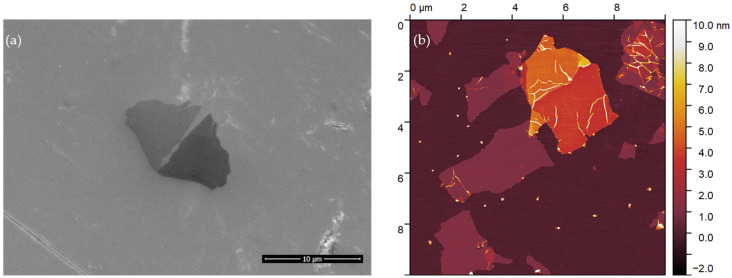
Image of graphene oxide (**a**) SEM magnification: 10000x; scale bar 10 µm; (**b**) AFM.

**Figure 2 ijms-24-12826-f002:**
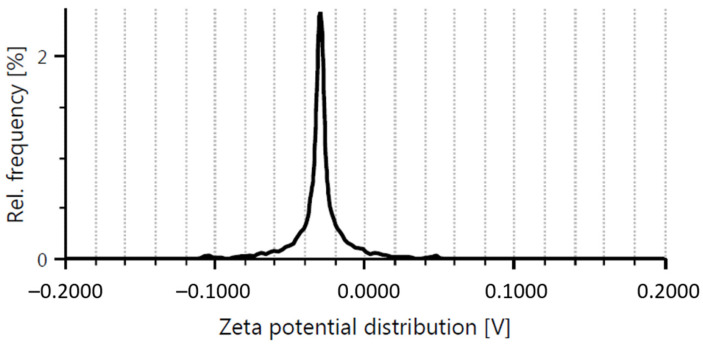
Zeta potential of GO dispersion in water.

**Figure 3 ijms-24-12826-f003:**
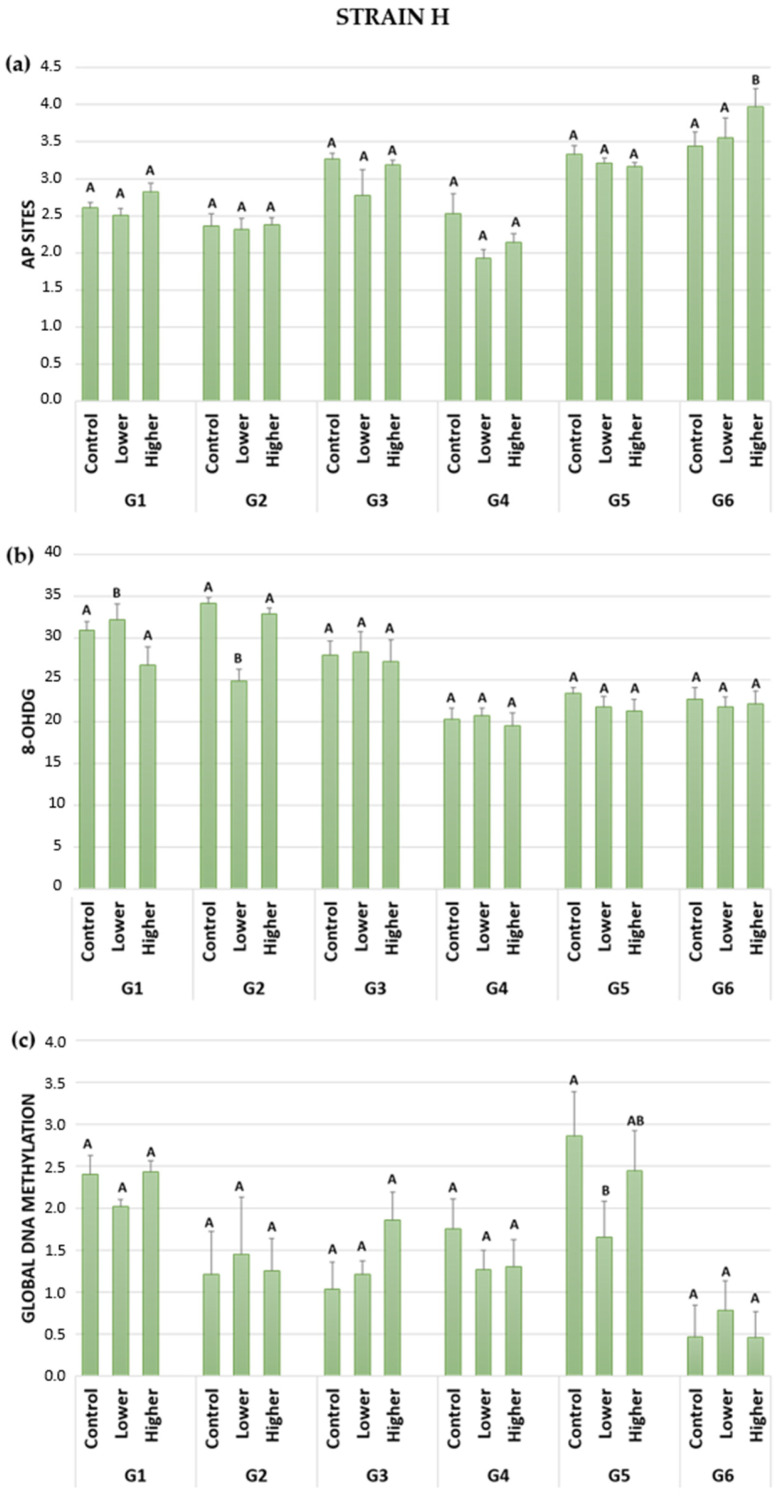
DNA stability parameters in the wild-type strain (H) of *A. domesticus* that had been chronically intoxicated with graphene oxide: (**a**) AP sites (apurinic/apyrimidinic sites); (**b**) 8-OHdG (8-hydroxy-2′-deoxyguanosine); (**c**) Global DNA methylation in the gut cells. Abbreviations: Generation 1–5 (G1–G5): Control animals fed uncontaminated food; lower and higher groups of animals fed GO-contaminated food at a concentration of 0.02 or 0.2 mg∙kg^−1^ of dry food, respectively; G6—animals fed uncontaminated food; n = 5; the data were normalized using Box–Cox transformation; significant differences were measured using ANOVA (Fisher test; *p* < 0.05); different letters denote differences among the experimental groups in the generation.

**Figure 4 ijms-24-12826-f004:**
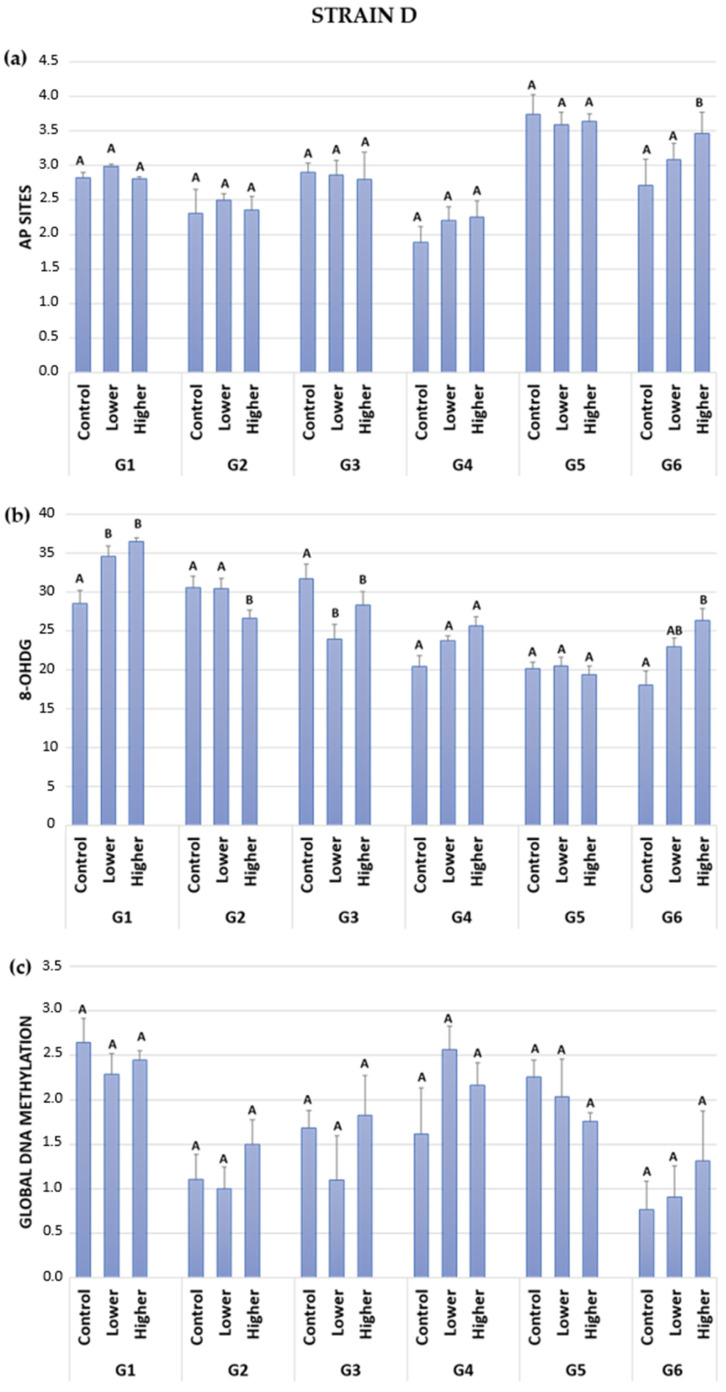
DNA stability parameters in the long-lived strain (D) of *A. domesticus* that had been chronically intoxicated with graphene oxide: (**a**) AP sites (apurinic/apyrimidinic sites), (**b**) 8-OHdG (8-hydroxy-2′-deoxyguanosine), (**c**) global DNA methylation in the gut cells of the long-living strain (D). Abbreviations: in Figure 3. Different letters denote differences among the experimental groups in the generation.

**Figure 5 ijms-24-12826-f005:**
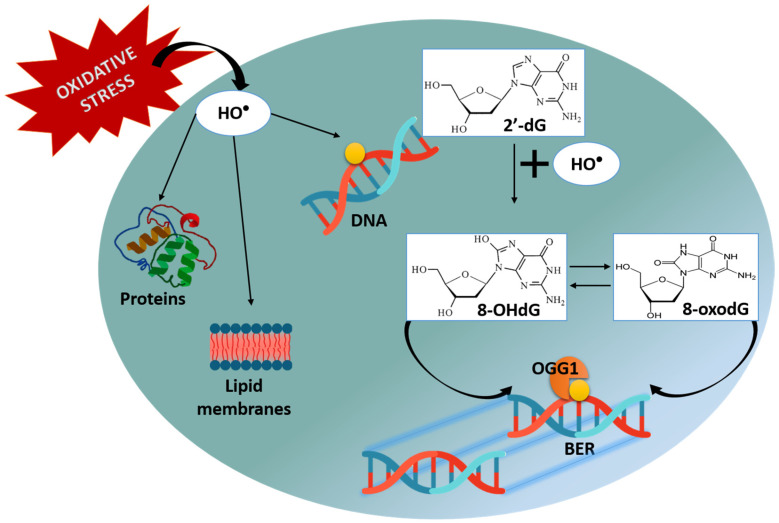
Simplified scheme of oxidative damage in the cell. Hydroxyl radicals can cause damage to proteins, membranes, and DNA. The HO· attacks DNA and oxidized products are formed. Mostly 2′-deoxyguanosine (2′-dG) in reaction with hydroxyl radical leads to the formation of 8-hydroxy-2′-deoxyguanosine (8-OHdG) or its tautomer 8-oxo-7-hydro-2′-deoxyguanosine (8-oxodG). Both changes can be promutagenic. In the process of Base Excision Repair (BER), the 8-oxoguanine glycosylase (OGG1) enzyme can cut out and exchange damaged bases.

**Figure 6 ijms-24-12826-f006:**
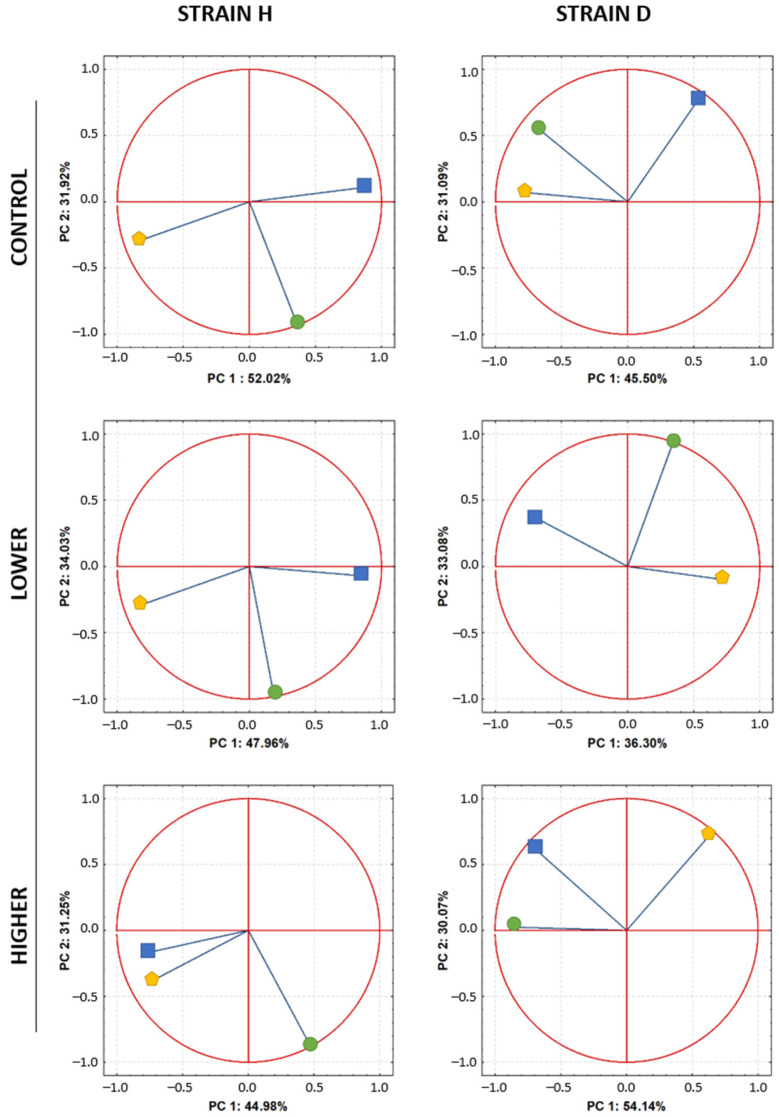
Principal component analysis (PCA) of the investigated DNA stability parameters: AP sites (pentagon), 8-OHdG (square), and global DNA methylation (circle) were analyzed in each strain and treatment group separately.

**Figure 7 ijms-24-12826-f007:**
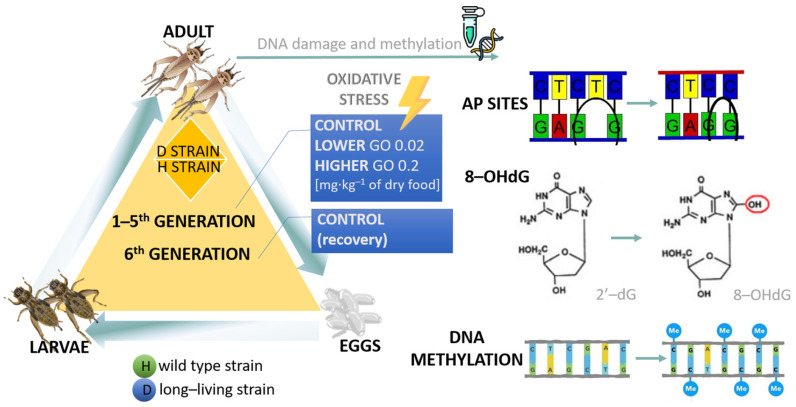
Scheme of the experimental model. See materials and methods (experimental model).

**Table 1 ijms-24-12826-t001:** The factor’s main effects and interactions of generation, treatment, and strain on investigated parameters. Aberrations: Effects—the factors (independent variables) and their interactions, i.e., mutual modifications of their impact on dependent variables. Tested variables: AP sites, 8-OHdG, global DNA methylation (MANOVA, Wilks’ Lambda test).

Effects	DNA Stability Parameters	
	F	*p*
Generation (G)	21.664	<0.0001
Treatment (T)	1.910	0.0791
Strain (S)	7.486	0.0002
G × T	3.307	<0.0001
T × S	2.502	0.0225
G × T × S	6.129	<0.0001

**Table 2 ijms-24-12826-t002:** Trends in (a) AP sites (apurinic/apyrimidinic sites), (b) 8-OHdG (8-hydroxy-2′-deoxyguanosine), and (c) global DNA methylation in the gut cells of the wild-type strain (H) of *A. domesticus* that had been chronically intoxicated with graphene oxide (GO). Abbreviations: as in Figure 3, the up/down arrows show the increase/decrease in the parameter level relative to the controls; the dashes show no differences in trends relative to the controls.

Strain H	Parameter	Treatment		
		Control	Lower	Higher
(a)	*AP sites*			
	G1	2.61	↓	↑
	G2	2.37	―	―
	G3	3.26	↓	↓
	G4	2.53	↓	↓
	G5	3.33	↓	↓
	G6	3.44	↑	↑
(b)	*8-OHdG*			
	G1	30.88	↑	↓
	G2	34.13	↓	↓
	G3	27.96	―	―
	G4	20.26	―	―
	G5	23.38	↓	↓
	G6	22.68	―	―
(c)	*Global DNA methylation*			
	G1	2.40	↓	―
	G2	1.21	↑	―
	G3	1.04	↑	↑
	G4	1.76	↓	↓
	G5	2.85	↓	↓
	G6	0.46	↑	―

**Table 3 ijms-24-12826-t003:** Trends in (a) AP sites (apurinic/apyrimidinic sites), (b) 8-OHdG (8-hydroxy-2′-deoxyguanosine), and (c) global DNA methylation in the gut cells of the long-living strain (D) of *A. domesticus* that had been chronically intoxicated with graphene oxide (GO). Abbreviations: as in Figure 3, the up/down arrows show the increase/decrease in the parameter level relative to the controls; the dashes show no differences in trends relative to the controls.

Strain D	Parameter	Treatment		
		Control	Lower	Higher
(a)	*AP sites*			
	G1	2.82	↑	―
	G2	2.31	↑	―
	G3	2.50	↓	↓
	G4	1.89	↑	↑
	G5	3.74	↓	↓
	G6	2.71	↑	↑
(b)	*8-OHdG*			
	G1	28.50	↑	↑
	G2	35.56	―	↓
	G3	31.68	↓	↓
	G4	20.52	↑	↑
	G5	20.14	―	―
	G6	18.00	↑	↑
(c)	*Global DNA methylation*			
	G1	2.65	↓	↓
	G2	1.10	↓	↑
	G3	1.68	↓	↑
	G4	1.61	↑	↑
	G5	2.25	↓	↓
	G6	0.76	↑	↑

## Data Availability

Raw data are provided on the RepOD database (accession: doi:10.18150/5C4X2O).

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
