# Peer review of "Multigenerational Effects of Graphene Oxide Nanoparticles on Acheta domesticus DNA Stability"

_ijms, 2023, doi:10.3390/ijms241612826_

Round 1

Reviewer 1 Report

2.2 Put a scheme of how DNA damage is carried out

no figure 2

The characteristics of graphene oxide are not part of materials and methods, they are part of the results, please change them.

Figure 4 corresponding to the SEM, this rare average, please improve it.

It would be necessary to carry out the XRD, BET and TEM analyzes and see how these studies can improve the conclusions of the authors.

Author Response

Dear Editor,

We are genuinely grateful for sending our manuscript to the reviewers. In the revised version of the manuscript, we have contained all the changes suggested by the reviewers. Detailed answers to the reviewers questions and a description of changes introduced in the manuscript are in “Response to Reviewers1.docx” and “Response to Reviewers2.docx” . Our comments and detailed descriptions of introduced changes according to the reviewers suggestions are given below.

Sincerely yours,

Maria Augustyniak, Barbara Flasz and co-authors

Reviewer #1:

Thank you for our manuscript's insightful, constructive, and helpful review. Thank you for pointing out our mistakes and significant matters that we omitted while preparing our manuscript. Once again, from a new perspective, we looked at our work. We did our best to address the issue better. Below we include detailed answers to your questions and explanations concerning your doubts.

Specific comments:

  1. 2 Put a scheme of how DNA damage is carried out.

Thank you for the suggestion. The scheme can easily explain the nature of oxidative DNA damage. We created the scheme of how DNA is damaged in the cell by oxidative stress, and we placed the figure (Figure 5) in the section “2.2. DNA damage: 8-OHdG (8-hydroxy-2’-deoxyguanosine)”. 

  1. no figure 2.

Thank you for that comment. In the manuscript, figures 1 and 2 were created to present DNA stability parameters in two investigated wild and long-lived strains. Figure 1 showed the parameters of AP sites, 8-OHdG, and DNA methylation in the wild type strain. These same parameters were presented in Figure 2, but in the long-lived strain that differs from the wild type in the ontogenetic development. Thus, both figures are necessary for the complete manuscript and complement each other. Perhaps you were misled by the insufficient description in Figure 2. This is why we have corrected this part of the manuscript. In the updated manuscript version, we have clarified the descriptions under Figures 1 and 2.

  1. The characteristics of graphene oxide are not part of the materials and methods, they are part of the results, please change them.

Thank you for a good point. In the corrected version of the manuscript, we present this paragraph rebuilt. We have changed that part of the manuscript and placed the characteristics of graphene oxide in the results part. The correction was made.

  1. Figure 4 corresponding to the SEM, this rare average, please improve it.

Thank you for a suggestion. The SEM picture was presented in the manuscript to illustrate the size and shape of GO flakes in two dimensions. At the same time, the AFM image showed information about the third dimension. We did our best to take pictures of the best possible quality. To meet Your requirements, we removed the previous picture, and in the corrected version of the manuscript, we placed the photo with a larger magnification.

  1. It would be necessary to carry out the XRD, BET and TEM analyzes and see how these studies can improve the conclusions of the authors.

Thank you for the valuable suggestion. You are right. The more analysis, the better. Using techniques like XRD, BET, TEM, and others is a good idea for our plans to give a bigger picture and more critical information that could fulfill the existing knowledge of nanoparticles’ mode of action, especially graphene oxide. We already have some experience in that field. In the past, we investigated four different graphene oxide nanoparticles that we bought from four companies. We did a very detailed analysis to gain as much as we can information about the physicochemical properties of GO. We also investigated the parameters of GO cytotoxicity. In detail, we presented GO characteristics using techniques: SEM, AFM, TEM, STEM-HAADF, SAED, HRTEM, Raman spectra, XPS, TOF-SIMS, and zeta potential. We measured cytotoxicity parameters like cell viability, oxidative stress, apoptosis, and comet assay. Due to the slightly different nature of this manuscript and the different range of results to be discussed, as well as considering the space available in the manuscript, we decided to present just a few parameters of graphene oxide characterization. Also, we quoted our earlier article and referred those interested to read it to get specific information about GO characteristics. Below You can find the information about the cited publication:

  • Dziewięcka, M., Pawlyta, M., Majchrzycki, Ł., Balin, K., Barteczko, S., Czerkawska, M., Augustyniak, M., 2021. The structure–properties–cytotoxicity interplay: A crucial pathway to determining graphene oxide biocompatibility. J. Mol. Sci. 22. https://doi.org/10.3390/ijms22105401

Reviewer 2 Report

1. The abstract starts a bit too far from the topic of the article - there is no need to describe what is graphene oxide in the abstract nowadays.

2. I do not understand what results the PCA analysis shows in this article - the results of the analysis should be either better clarified and described in more details or this paragraph should be removed.

3. What was the reason to use this group size in the experiments and can the larger groups improve the results?

4. Raman spectra or Kelvin mode in AFM can provide slightly more information about state of the graphene oxide (GO) flake, but the already provided characterization is generally enough to understand the GO state.

5. Size distribution of the flakes can affect some metabolic properties of the GO, is there some analysis about size distribution of the particles.

There is no problems with the language.

Author Response

Dear Editor,

We are genuinely grateful for sending our manuscript to the reviewers. In the revised version of the manuscript, we have contained all the changes suggested by the reviewers. Detailed answers to the reviewers questions and a description of changes introduced in the manuscript are in “Response to Reviewers1.docx” and “Response to Reviewers2.docx”. Our comments and detailed descriptions of introduced changes according to the reviewers suggestions are given below.

Sincerely yours,

Maria Augustyniak, Barbara Flasz and co-authors

Reviewer #2:

Thank you for your kind reviews, valuable remarks, and suggestions. They were beneficial during the preparation of the corrected version of our manuscript. All of them have been considered, and considerable and satisfying changes were introduced into the updated version of the manuscript. We did our best to improve the manuscript. We have followed your suggestions. We have improved the abstract and answered all of Your questions.  Due to the changes, we hope the revised manuscript is more valuable. Below are our replies to your points.

Specific comments:

  1. The abstract starts a bit too far from the topic of the article - there is no need to describe what is graphene oxide in the abstract nowadays.

Thank You for that valuable suggestion. We changed the abstract and rebuilt it. There is no more unnecessary information about graphene oxide, and we hope the abstract is now more concise, readable, and generally improved.

  1. I do not understand what results the PCA analysis shows in this article - the results of the analysis should be either better clarified and described in more details or this paragraph should be removed.

Thank you for your good point. We used Principal Component Analysis to show general relationships and regularities between all variables. It does not require the creation of preliminary hypotheses. During the analysis, the variables are reduced to two principal components to explain the variability of the initial data as much as possible. This type of analysis can only show us that individual variables are somehow related to each other. It may reveal some hidden dependencies. However, it is difficult to conclude on the basis of why the variables are correlated. We want to avoid removing the graph from the manuscript because PCA presents essential information about dependencies between investigated groups based on correlation. The PCA is a good tool in that case to investigate the differences. We used it to replace the correlation, which allows you to show the relationship between only two variables, to a comprehensive study of relationships and visualize (global) possible relationships. We tried to meet your requirements, and we have improved the description of (PCA) as much as possible in the corrected version of the manuscript.

  1. What was the reason for using this group size in the experiments and can the larger groups improve the results?

Thank You for that question. The larger groups, the better results. If there are more repeats, we can get more accurate and precise results with lower standard deviation. In an ideal arrangement, many repetitions would improve the results. But in traditional experiment planning, we follow the principle of the 3Rs. The 3Rs principle is commonly used in animal research. The 3Rs stand for replacement, reduction, and refinement.  The replacement means methods that avoid or replace the use of animals in research. Reduction is using techniques that enable obtaining comparable levels of information from fewer animals or more information from that same number of tested animals. Refinement means using methods that can minimize pain, suffering, and distress and enhance animal welfare for the animals used. According to the 3Rs principles, we believe a trade-off between accuracy and the number of animals sacrificed must be maintained. On the other hand, with large group sizes and many repetitions, there is an economic problem. A large number of tested individuals is associated with large consumption of reagents and, therefore, high costs. Also, large groups can be problematic because of logistic problems in the breeding rooms. Thank You for understanding our point of view.

  1. Raman spectra or Kelvin mode in AFM can provide slightly more information about state of the graphene oxide (GO) flake, but the already provided characterization is generally enough to understand the GO state.

Thank You for that suggestion. We agree with You. The more analysis, the better. In our previous publication, we presented the physiochemical characteristics of four different graphene oxide suspensions from various companies in detail. We investigated GO flakes using many techniques (SEM, AFM, TEM, STEM-HAADF, SAED, HRTEM, Raman spectra, XPS, TOF-SIMS, and zeta potential), and we measured some parameters of cell health status like cell viability, oxidative stress, apoptosis, comet assay. We did it to assess how different the nanoparticles supplied by other companies are and whether their potential differences in physicochemical properties may affect cytotoxicity. As You suggested in the comment, the GO characteristic is sufficient for this manuscript, and we agree. Below You can find detailed information about our previous work concerning a detailed study on GO:

  • Dziewięcka, M., Pawlyta, M., Majchrzycki, Ł., Balin, K., Barteczko, S., Czerkawska, M., Augustyniak, M., 2021. The structure–properties–cytotoxicity interplay: A crucial pathway to determining graphene oxide biocompatibility. Int. J. Mol. Sci. 22. https://doi.org/10.3390/ijms22105401

  1. Size distribution of the flakes can affect some metabolic properties of the GO, is there some analysis about size distribution of the particles.

Thank You for that valuable comment. The size distribution of particles is very interesting and worth to explore the topic. However, this requires a separate material analysis study and requires analyzing multiple samples. It also involves much time. Analysis of size distribution was carried out in the mentioned publication (Dziewięcka et al., 2021). In this research, we have used graphene oxide, investigated in the work of Dziewięcka et al. (2021), as sample number 3. Detailed characteristics of GO used in that research are available in the mentioned publication.  We merely refer to this article in the manuscript without wishing to duplicate the results. The flake size analysis was conducted on more than 400 flakes in the sample. The average flake area is about 2 µm2. In the corrected manuscript version, we added that important information to the GO characteristics part in the results section.

  1. There is no problems with the language.

Thank You for that comment.

Round 2

Reviewer 1 Report

Now, this manuscript could be accept in present form.

Reviewer 2 Report

The authors answered most of concerns about the original draft of the article and the article can be accepted in present form. I still have my opinion that the PCA analysis does not add clarity to the article and can be omitted or replaced with some other method, but I think it can be left as is after this revision.